# Chemically Functionalized Cellulose Nanocrystals as Reactive Filler in Bio-Based Polyurethane Foams

**DOI:** 10.3390/polym13152556

**Published:** 2021-07-31

**Authors:** Francesca Coccia, Liudmyla Gryshchuk, Pierluigi Moimare, Ferdinando de Luca Bossa, Chiara Santillo, Einav Barak-Kulbak, Letizia Verdolotti, Laura Boggioni, Giuseppe Cesare Lama

**Affiliations:** 1Institute of Chemical Science and Technologies—“G. Natta”, National Research Council, via A. Corti 12, 20133 Milan, Italy; francesca.coccia@cnr.it (F.C.); pierluigi.moimare@cnr.it (P.M.); 2Leibniz-Institut für Verbundwerkstoffe GmbH, Technische Universität, Erwin-Schrödinger-Straße 58, 67663 Kaiserslautern, Germany; Liudmyla.Gryshchuk@ivw.uni-kl.de; 3Institute of Polymers, Composite and Biomaterials, National Research Council, Piazzale Enrico Fermi, 80055 Portici, Italy; fe.delucabossa@gmail.com (F.d.L.B.); chiara.santillo@cnr.it (C.S.); giuseppe.lama@ipcb.cnr.it (G.C.L.); 4Melodea Ltd., Rehovot 7610001, Israel; einav.kulbak@melodea.eu

**Keywords:** reactive filler, cellulose nanocrystals, bio-based polymers, polyurethane foams

## Abstract

Cellulose Nanocrystals, CNC, opportunely functionalized are proposed as reactive fillers in bio-based flexible polyurethane foams to improve, mainly, their mechanical properties. To overcome the cellulose hydrophilicity, CNC was functionalized on its surface by linking covalently a suitable bio-based polyol to obtain a grafted-CNC. The polyols grafted with CNC will react with the isocyanate in the preparation of the polyurethane foams. An attractive way to introduce functionalities on cellulose surfaces in aqueous media is silane chemistry by using functional trialkoxy silanes, X-Si (OR)_3_. Here, we report the synthesis of CNC-grafted-biopolyol to be used as a successful reactive filler in bio-based polyurethane foams, PUFs. The alkyl silanes were used as efficient coupling agents for the grafting of CNC and bio-polyols. Four strategies to obtain CNC-grafted-polyol were fine-tuned to use CNC as an active filler in PUFs. The effective grafting of the bio polyol on CNC was evaluated by FTIR analysis, and the amount of grafted polyol by thermogravimetric analysis. Finally, the morphological, thermal and mechanical properties and hydrophobicity of filled PUFs were thoughtfully assessed as well as the structure of the foams and, in particular, of the edges and walls of the cell foams by means of the Gibson–Ashby model. Improved thermal stability and mechanical properties of PU foams containing CNC-functionalized-polyol are observed. The morphology of the PU foams is also influenced by the functionalization of the CNC.

## 1. Introduction

Cellulose is one of the most abundant bio-polymers on Earth. It represents 40–60% of wood mass, can be extracted in the form of 20–40 µm thick fibers upon pulping [1] and is processed into functional nanoparticles. Among these, cellulose nanocrystals (CNC) are crystalline nano-rods with a diameter of 3–10 nm and a length of a few hundred nanometers. They are extracted from pulp fibers using an acid-mediated procedure [2] and can be utilized for the enhancement of novel eco-friendly materials [3] such as polyurethane foams (PUs) [4,5,6]. 

PUs are generally composed of polyol, isocyanate and several additives. Pus’ mechanical and functional properties are determined by the typology of polyols and isocyanate precursors, by the consequent distribution of the soft and hard segments that composed the PUs polymeric matrix and by the morphological structure in terms of open or closed cells [7]. Nowadays, the feasibility of using bio-based polyols to produce PUs has been strongly investigated due to the wide application fields accessible for foamed products such as biomedical, construction, aerospace, etc. Among bio-based polyols, the Mannich-based polyols derived from Cashew Nut Shell Liquid (CNSL) [8,9,10], due to their chemical structure, phenolic ring and a tertiary amine (Scheme 1), can be used for the synthesis of bio-based polyurethane (bio-PUs). 

The presence of a filler, such as CNC, could act as heterogeneous nuclei for crystallization of PU and, consequently, increase the mechanical properties of foam [11,12,13]. The combination of cellulose characteristics, such as low cost, chemical modification capacity, high Young’s modulus, biodegradability, abundance in nature, along with features of the nanosized materials, such as very large specific surface area, high aspect ratio, light weight, and outstanding reinforcing potential, justifies the big interest of scientists in CNC [14]. On the other hand, the cellulose hydrophilicity leads to a bad dispersion of non-polar matrices, and the consequential nanocrystal aggregation could decrease thermo-physical and mechanical properties of the nanocomposites [15]. The reinforcing mechanism, in fact, depends mainly on the formation of a three-dimensional network between the filler surface and matrix. Such interaction can also be improved by the formation of strong interfacial interactions or by a co-continuous phase mediated by grafted polymer chains on CNC surface [16,17]. 

Physical and chemical modification of CNC [18] plays a key role in improving dispersibility in solvent, regulating miscibility with the hydrophobic polymer matrix, and enhancing thermal stability [19,20]. Functionalized PLA nanocellulose composites show better properties than their unmodified counterparts due to improved dispersion and interaction with the PLA matrix [21].

The abundant hydroxyl groups on CNC are an essential route to altering the surface structure, regulating properties and developing functional materials. A wide range of chemical modifications can be placed on CNC surfaces through simple reactions such as oxidation and acetylation [22,23] and sometimes, the modification involved the grafting of functional materials and polymers. 

In the literature, few examples of the formation of modified cellulose by hydrophobic polyols are reported. In 2019, Xu et al. [24] grafted cellulose pulp with polymeric epoxidized soybean oil (ESO), as a renewable environmentally friendly and low-cost raw material, with the aim to prepare oil-absorbing materials. In the same year, Gorade et al. [25] achieved microcrystalline cellulose (MCC) with hydrophobic property by means of chemical transesterification using rice bran oil (RBO). At the same time, Alzate-Arbeláez et al. [26] loaded nanocellulose, derived from banana rachis, with a polyphenolic-rich extract of Andean berries (Vaccinium meridionale) through simple impregnation obtaining an antioxidant nanocomplex. Instead, bio-based formulations with adhesion properties were synthesized first by inducing the functionalization of cellulose acetate with 1,6-hexamethylene diisocyanate and then mixing the resulting biopolymer with a variable amount of castor oil to obtain bio-based polyurethane adhesives showing more suitable mechanical properties [27].

A typical CNC monomeric unit presents at least three reactive groups suitable to be functionalized, but for sterical reasons, only one is available. Otherwise, polyols contain many hydroxyl groups, whose bulky moieties prevent the direct reaction with the reactive group of CNC. For this reason, it is necessary to introduce a spatial linker, such as a silane, to make the reaction happen. An attractive method to functionalize cellulose surface in aqueous media is by using silane chemistry. Suitable substituted trialkoxy silanes (X-Si (OR)_3_), with vinyl, thiol and azido groups, can be used to impart to cellulose reactive moieties. Azido silanes are often used to promote further functionalization through click reactions [28,29]. 3-mercaptopryltrimethoxysilane is used to introduce -SH functionality on cellulose by thiol-ene reaction [30]. The major problem regarding the silanization is related to the hornification that causes pore system collapse and results in a decrease in reactivity and in a modification in the 3D structure of the material. The collapse is attributed to the modification conditions which may induce disorder and misalignment of the structure of cellulose fibers (e.g., axial orientation of molecular chains and crystalline phase of the fiber are reduced) [31]. However, silanization is classified as a mild and green approach for cellulose modification because it could occur in an aqueous medium [32]. In contrast to conventional methods, no curing or solvent exchange is necessary, thereby the functionalized celluloses remain never-dried, and no agglomeration or hornification occurs in the process [33]. Often the cellulose functionalization is used to increase the hydrophobic character of cellulose nanofibrils. Fathi et al. in 2017 observed that cellulose TEMPO oxidation significantly improved the bonding efficiency of the silane coupling agents on the flax fibers surface thus, the compatibility between the flax fibers and the polymer matrices, such as epoxy resin, was improved [34]. Zimmermann et al. in 2016 modified cellulose fibers with triethoxyvinylsilane to be used as reinforcement agents in poly (ethylene-vinyl acetate) composites. The samples showed greater stiffness and increased mechanical properties creating a more compatible interface with the polymeric matrix [35]. Rachini et al. in 2008 asserted that the formation of a chemical bond between the hydroxyl groups of the hemp fibers and the hydrolyzed silane molecules improves even the thermal decomposition of the hemp fibers [36].

In this paper, for the first time, we report the new approach of functionalization of CNC with bio-polyol, through the silane chemistry functionalization, in order to be used as a successful reactive filler in bio-based polyurethane flexible foams, PUFs. The alkyl silanes were used as efficient coupling agents for the grafting of CNC and bio-polyols. Four different strategies to obtain CNC grafted with polyol (CNC-grafted-Polyol) were fine-tuned to use CNC as reactive filler in bio-based PUFs (Scheme 1).

The effective grafting of the bio polyol on CNC was evaluated by FTIR analysis, and the amount of grafted polyol by thermogravimetric analysis (TGA). 

Finally, the morphological and mechanical properties and hydrophobicity of filled PUs were thoughtfully assessed as well as the structure of the foams and in particular of the edges and walls of the cell foams by means of the modified Gibson–Ashby model [37,38,39].

## 2. Materials and Methods

### 2.1. Materials 

The bio-polyol GX9104, a cashew nutshell liquid (CNSL) based polyol with an OH value of 245 mg KOH/1 g oil, was supplied by Cardolite Corporation (Bristol, PA, USA). The bio-polyol RV33, a polyol with an OH value of 78 mg KOH/g, was supplied by AEP Polymers S.r.l. (Basovizza, TS, Italy). Sovermol^®^ 750, a bio-polyol with an OH value of value 300–330 mg KOH/g, was purchased by BASF. Sovermol^®^ 815, a bio-polyol with an OH value of 200–230 mg KOH/g, was purchased by BASF. Glycerol was purchased by Sigma-Aldrich (St. Louis, MO, USA). The co-reactive flame retardant Exolit^®^ OP 560 with an OH value of 450 mg KOH/g was purchased by Clariant. Blowing catalyst Dabco^®^ 33-LV and cross-linking Dabco^®^ NE 300 were purchased from Evonik. Silicone surfactant TEGOSTAB^®^ B 8747 LF2 was purchased from Evonik. Lupranate M20S poly(4,4′-Diphenylmethandiisocyanat) with 31.5% NCO-content was purchased from BASF.

The CNC used was supplied by MELODEA Ltd. (Rehovot, Israel) as a 3% aqueous suspension. In Table 1 we report the characteristics of CNC used in the experiments.

The TEA (triethylamine) was purchase from Sigma-Aldrich (St. Louis, MO, USA) and distilled over calcium hydride before use. The silanes (trimethoxy(propyl)silane, trimethoxy(octyl)silane, trichloro(propyl)silane and (3-Aminopropyl) triethoxysilane (APTES) and the solvents used (Dimethylformamide (DMF) dry, Tetrahydrofuran (THF), Methanol, Ethanol, Acetone) are purchased from Sigma-Aldrich (St. Louis, MO, USA) and used without any purification.

### 2.2. Modified *CNC* Preparations

#### 2.2.1. Dried Lyophilized CNC Preparation 

The dried lyophilized CNC powder was obtained by using the liophilizer LIO-5P, (Cinquepascal S.R.L, Trezzano Sul Naviglio, Italy) and the ball milling Retsch MM 400 (Retsch GmbH, Haan, Germany). 

#### 2.2.2. CNC_grafted_TMPS_polyol (CNC1) and CNC_grafted_TMOS_polyol (CNC2) Preparations

Trimethoxy(propyl)silane (1 eq, 1.2 mL, 1.12 g) (TMPS) or trimethoxy(octyl)silane (1 eq, 1.8 mL, 1.63 g) (TMOS) was hydrolyzed in acid aqueous solution (HCl solution 0.5 M, 0.5 eq) for 10 min at room temperature, then the GX9104 polyol (1 eq, OH value = 245 mg KOH/g, *M*_w_ = 2886 g/mol), in THF solution (20 mL), was added and the mixture kept at 70 °C for 20 min. Then, 14 mL of NaOH (0.5 M, 0.5 eq) was added and the solution warmed at 80 °C for 4 h. 

CNC as a 3 wt% aqueous suspension (1 eq, 36.6 mL) was added and the reaction was left overnight at 100 °C. The mixture was washed using THF, methanol and finally acetone, respectively. During each washing step, the mixture was sonicated for 15 min in an ultrasound bath and then centrifuged at 6000 RPM. The solid recovered was dried in an oven at 80 °C under vacuum (100 mbar) for 1 night. Finally, the sample was pre-freezed in liquid nitrogen and then pulverized in a ball milling for 2 min using a frequency of 30 oscillations/sec until a white powder was obtained (Scheme 2).

#### 2.2.3. CNC_grafted_TCPS_polyol (CNC3) Preparations

2 g (1 eq) of lyophilized CNC was suspended in a nitrogen atmosphere in 70 mL of dry DMF at 120 °C for 4 h, then the reaction was cooled and trichloro(propyl)silane (TCPS) added (1 equiv, 2 mL, 2.39 g) together with TEA (1 equiv) and left to react for 2 h at 60 °C. After that, a further amount of TEA (2 mL) was added together with the polyol GX9104 (1 equiv), previously dissolved in 4 mL of dry DMF, and the mixture was left overnight at 100 °C. After that, the mixture was washed using THF, methanol and finally acetone, respectively. During each washing step, the mixture was sonicated for 15 min in an ultrasound bath and then centrifuged at 6000 RPM. The solid recovered was dried in an oven at 80 °C under vacuum (100 mbar) for 1 night. Finally, the sample was pre-freezed in liquid nitrogen and then pulverized in a ball milling for 2 min using a frequency of 30 oscillations/sec until a white powder was obtained (Scheme 3).

#### 2.2.4. CNC_grafted_APTES_polyol (CNC4) Preparations

In a beaker, 5 equiv of APTES was solubilized in a solution of water (190 mL) and ethanol (190 mL). Acetic acid was dropped to have a pH = 5, then 108 mL of CNC 3 wt% suspension (1 equiv) was added and keep for 2 h at room temperature. The mixture was then filtered and put in a 250 mL round-bottomed flask at 110 °C for 1 h. 

Then, 1 eq of GX9104 polyol was mixed with 3 eq of formic acid (2.3 mL) for 15 min at room temperature, and 7.5 equiv of H_2_O_2_ was added drop by drop and then the solution was left overnight at 50 °C. After that, an aqueous solution of NaHCO_3_ was added and the mixture was washed with CHCl_3_ several times; then, the organic phase was dried over NaSO_4_ and the solvent was removed under reduced pressure. The obtained epoxidized polyol was analyzed by FT-IR (Spectrometer Perkin Elmer, Waltham, Massachusetts, U.S.) and the disappearance of the signal at 3007 cm^−1^, typical of double bound, was observed as well as the appearance of the characteristic features of the epoxy group at 823, 871 and 917 cm^−1^.

The epoxidized polyol was dissolved in THF (0.6 eq) and mixed with CNC silylated with APTES. The reaction was left at 100 °C for 3 days. The product obtained was analyzed by FT-IR spectrometer (Scheme 4).

#### 2.2.5. CNC Purification

All the modified CNC prepared, CNC1, CNC2, CNC3 and CNC4, were purified by unreacted polyols and/or silanes by using a Soxhlet apparatus and THF as solvent. 

### 2.3. PU Foams Preparation

The bio-based polyurethane (PU) foams were prepared by using a one-step method. *Component A* was prepared by mixing in a plastic beaker with a mechanical stirrer, for 5 min at 500 rpm the following bio-based polyols: RV33, Cardolite^®^ NX-9201 (polyester diol), Sovermol^®^ 750 (branched polyether/polyester polyol), Sovermol^®^ 815 (branched polyether/polyester polyol), glycerol as chain extender, blowing agent water, co-reactive flame retardant Exolit^®^ OP 560, blowing and cross-linking catalysts Dabco^®^ 33-LV and Dabco^®^ NE 300 respectively, silicone surfactant TEGOSTAB^®^ B 8747 LF2. 

An appropriate amount ([NCO]/[OH] = 1.05) of Component B (poly(4,4′-Diphenylmethandiisocyanate) Lupranate M20S was added to Component A and the mixture was stirred with a mechanical stirrer for 20 s at 500 rpm. Afterward, immediately, the resultant mixture was transferred to an open cylindrical mold and allowed to free rising at room temperature. In Table 2, the amounts of reagents for the PU formulation are reported.

For the preparation of filled bio-based PU foams, an according amount of filler was mixed with Component A at the mass ratio of 1, 2 and 3 wt% and the mixture was stirred for 5 min at 500 rpm. Then, the isocyanate was added following the same procedure described for the non-filled foam (namely PU-Ref). The produced composite panels were labeled PU-CNCx_y, where x is the number associated with the selected filler (can be equal to 1, 2, 3 or 4), and y is the amount of filler dispersed within the matrix (can be equal to 1, 2 or 3)

### 2.4. Characterization of Modified CNCs and PU Foams

#### 2.4.1. Modified CNCs Characterization

All the modified CNC prepared, CNC1, CNC2, CNC3 and CNC4, were characterized by FT-IR Spectrometer Perkin Elmer (Waltham, MA, U.S.) in Attenuated Total Reflectance (ATR) mode from 400−4000 cm^−1^, 4 cm^−1^ resolutions and 64 scans. The spectra were normalized between 0 to 100 on transmittance value.

Structural characterization of CNCs was performed through Wide Angle X-ray Scattering (WAXS) analysis using an Anton Paar SAXSess camera equipped with a 2D imaging plate detector. Then, 1.5418 Å wavelengths CuKα X-Rays were generated by a Philips PW3830 sealed tube generator source (40 kV, 50 mA) and slit collimated. Spectra of CNCs powder were collected for 10 min. The crystallinity index (*CrI*, %) was determined using the empirical method, known as the Segal or peak height method, [40] as follows:(1)$$CrI=\frac{I_{200}-I_{am}}{I_{am}}\times 100$$
where *CrI* expresses the relative degree of crystallinity, *I*_200_ is the maximum intensity (in arbitrary units) of the 200 lattice diffraction and *I_am_* is the intensity of diffraction in the same units at 2θ = 18°.

Thermo-gravimetric (TGA) analyses, by using a TGA 7 Perkin Elmer (Waltham, MA, USA) over a temperature range from 30 °C to 800 °C under nitrogen atmosphere, were performed on modified CNCs to evaluate the amount of silane and polyols grafted (see Equation (7)).

#### 2.4.2. PU Foams Characterization

FTIR spectra of foamed samples were collected at room temperature by using an FT-IR spectrometer (Bruker IFS 66 FT-IR equipment) in attenuated total reflectance (ATR) mode from 400−4000 cm^−1^, 4 cm^−1^ resolutions and 64 scans.

The morphological structure of polyurethane foams was investigated by scanning electron microscopy (SEM, FEI ESEM Quanta 200, acceleration voltage of 15 or 20 kV). The density of PU foams was determined at 23 °C with 50% relative humidity (ASTM D1622-03). The density value reported is the average value of 10 specimens with size 30 mm × 30 mm × 30 mm (length × width × thickness).

Hydrophobicity of filled PU foams was determined by optical contact angle analysis instrument OCA 25 (DataPhysics Instruments GmbH, Berlin, Germany). The instrument measures the contact angle of a drop of water placed by a dosing needle on the foam surface. The value of contact angle is the average value of 10 tests.

Mechanical compressive strength of PU foams was determined according to DIN EN ISO 844 and carried out through Zwick 1445 Retroline machine (ZwickRoell GmbH and Co. KG, Berlin, Germany). The following parameters were used for measurement: initial load 0.5 N, E-modulus velocity 10 mm/min, testing velocity 10%/min, maximal deformation 70%. Compressive strength at 10% and 40% strain and according to values of compressive modulus were performed. Then, 10 specimens were tested and an average value was taken along with the standard deviation.

TGA/DTG analysis of PU foams was carried out with TGA/DSC 3+ STAReSystem (Mettler Toledo, Berlin, Germany). An amount of 5–10 mg of powder sample was loaded in the aluminum crucible and analyzed by the thermal analyzer. Thermal degradation was terminated from RT up to 580 °C with a heating rate of 10 °C/min under an air flow of 50 mL/min.

#### 2.4.3. Modified Gibson–Ashby Model

Even though the foam cell structure can be appreciated from the SEM analysis, the direct correlation between the cell walls or edges structure and mechanical properties is better elucidated by the Gibson–Ashby model [37,38,39]. It correlates the relative density (density of the foam over the density of the cell-walls (or cell-edges) material, ρ/ρ_s_) with the mechanical properties of the foam, such as the relative Young’s modulus given by the ratio between the elastic modulus of the foamed material (E) and the elastic modulus of the material constituting the cell walls (Es) by Equation (2).
(2)$$\left(\frac{E}{E_{s}}\right)=C\phi^{2}{\left(\frac{\rho }{\rho_{S}}\right)}^{2}+C^{\prime }\left(1-\phi \right)\left(\frac{\rho }{\rho_{s}}\right)$$
where *C* and *C*′ are usually equal to 1, and the factor ϕ is the fraction of volume condensate in the cell edges. This equation is usually related to foams with closed-cell structures. If ϕ = 1, then the equation becomes
(3)$$\left(\frac{E}{E_{s}}\right)=C{\left(\frac{\rho }{\rho_{S}}\right)}^{2}$$
which models open-cell foams. However, a more generic equation could explain how the pores constituting the foam are made:(4)$$\left(\frac{E}{E_{s}}\right)=C{\left(\frac{\rho }{\rho_{S}}\right)}^{n}$$
when *C* = 1, it is possible to obtain the value of the exponent *n* for each set of data.

Here we propose a modified Gibson–Ashby equation to model the linear elasticity of our PU foams with a hybrid open/closed cells structure. We start with Equation (2) to obtain ϕ values for each foam. In this way, we collected a dataset of the average pore of the single PU foam is constituted, in terms of material distribution between edges and wall.

Once the dataset was collected, we used it to develop the modified Gibson–Ashby model that could better fit with our outcomes.
(5)$$\left(\frac{E}{E_{s}}\right)=C\prime \prime \left[\phi^{2}+\left(\frac{\rho }{\rho_{s}}\right)\right]$$
(6)$$C^{\prime \prime }=a-\phi^{\frac{n}{b}}$$
with *C*″ (given by Equation (6)) is a constant, depending on the variables (ϕ and *n*) by a power law, by means of two arbitrary constants *a* and *b*. Since the equation is non-linear, it is solved by successive attempts, in order to obtain a general value of *C*″, from which each out-coming best fit does not differ from the relative modulus and relative density of the corresponding material of more than 1%. In our specific case, the two constants resulted to be *a* = 1.7 and *b* = 5.

Substituting the exponent *n* with the values found with Equation (4) and the factor ϕ with values found with Equation (2), the presented model fit the measured mechanical parameters best and was better at elucidating the structure of the foam.

## 3. Results and Discussion

### 3.1. Characterization of *CNC*s-Grafted-Silanes-Polyol

#### 3.1.1. FTIR Characterization

The occurred grafting of all the silanes and polyol on CNC, after Soxhlet extraction (see Section 2.2 in Experimental details), was confirmed by FTIR analysis. Figure 1 shows the FTIR spectra of pristine CNC and the functionalized CNC’s with silane and polyol. 

In Figure 2, three enlargements of Figure 1 are shown.

The OH groups appear as broad adsorption peaks between 3600–3100 cm^−1^ and a slight decrease in intensity was observed by comparing the neat CNC and the functionalized one. In the spectral region between 3000–2700 cm^−1^ (Figure 2a) there are appreciable differences for the signals of neat and functionalized CNC corresponding to CH and CH_2_ asymmetric stretching of the cellulose, silane and polyol.

In the region between 1870 and 1600 cm^−1^ (Figure 2b), the carbonyl peak of polyol appears at 1718 cm^−1^ for the CNC2, CNC3 and CNC4, while for the CNC1, it is slightly shifted to 1757 cm^−1^. This peak, corresponding to the carbonyl of the polyol moiety, is a clear indication of the functionalization of the cellulose. Furthermore, this signal is completely absent for the neat CNC.

The main difference in the region between 1270–451 cm^−1^ (Figure 2c) is the shift of the absorption peak from 810 to 800 cm^−1^ of Si-C and Si-O stretching [31] of the neat CNC.

No other significant differences were observed in these regions.

#### 3.1.2. TGA Analysis

The obtained CNC1, CNC2 CNC3 and CNC4 products were analyzed by TGA to determine the residue and the results compared with the neat CNC. Samples were heated from 30 °C to 800 °C under nitrogen flow (Figure 3). Table 3 reports the temperatures for the 5, 10 and 50% of weight loss and the percentage of residue for all the samples taken into consideration.

TGA analysis was used, as proposed by Fathi [34] and Rachini [36], to determine the amount of silane on the CNCs. In fact, by correlating the final residue of the neat and grafted CNC after 400 °C, which corresponds to the full degradation of cellulose, it is possible to calculate the quantity of grafted silanes on the CNC surface.
Grafted (silane) (mmol/g of CNC) = [*W*_150–400_**/**(*M*_silane hydrolyzed_)] × 1000(7)
where *W*150-400 is the difference (%) of the weight loss in the presence and in the absence of the CNC and M the molecular mass of the hydrolyzed silane. The results are reported in Table 4. 

#### 3.1.3. WAXS Characterization of CNCs-Grafted-Silanes

Figure 4 shows the structural phases present in pristine CNC and CNCs*-grafted-*silanes samples. It is well-known that pristine CNC consists of chains that are arranged parallel to one another and are held together by strong H bonds, forming fibrils. These fibrils are locally orderly, to the extent of having a crystalline structure (namely type IV_1_) [41] observed by analyzing the diffractogram in Figure 4a which shows a typical pattern of cellulose I, with two main peaks at 2θ around 15.3° correlating to the overlapping of the reflection planes (1$\overset{¯}{1}$0) and (110), and 2θ equal to 22.1° due to the reflection plane (200) [42]. However, a shoulder at 2θ equal to 20.3° on the lower side of the (200) plane is also evident. This shoulder can be associated with the reflection plane (110) of cellulose II [43] and indicates that the pristine CNC crystallizes into a mixture of Form I and II. The diffraction patterns of grafted-CNCs reveal that the cellulose crystallizes in a mixture of the two forms even after functionalization. The addition of the silane component does not significantly affect the amount of crystallinity of the CNC; in fact, crystallinity indices (CrI) of 55%, 56% and 58% were calculated for pristine CNC, CNC_TCPS and CNC_TMPS respectively, while a modification of crystallinity in terms of shift of 2θ is observed, for instance, the peak at 22° shift to higher 2θ for both CNC_TMPS and CNC_TCPS. Furthermore, for the CNC_TCPS, a disappearance of the peak at 15° is recorded. This behavior could be correlated to the chemical interaction between the silane and the OH of cellulose and in detail to the intercalation of silane within the fibril structure of cellulose as schematized in Figure 4b. In particular, the intercalation is higher for TCPS with respect to the more sterically hindered TMPS and to more reactivity of chloride silane. This promotes a partial exfoliation of the fibrils, which became available, through OH groups, to react with isocyanate groups. However, the presence of the peak at 2θ value of 4.6° and 7.4° for CNC_TCPS and CNC_TMPS respectively can be related to the (001) basal spacing of the silane component which tends to interact with itself rather than with cellulose [44].

### 3.2. PUFs Characterization

#### 3.2.1. FTIR Characterization

FTIR spectra related to the selected polyurethane foam composites (PU-CNC_3, PU-CNC1_1, PU-CNC3_3) and the PU reference are reported in Figure 5, in which the main characteristic region of urethane groups (2000–800 cm^−1^) is shown. By analyzing the spectra, the absence of the signal at 2270 cm^−1^, related to the asymmetric stretching of the free NCO group, is detected [45]. Furthermore, in the FTIR spectra, the vibration peak at 1710 cm^−1^ related to the carbonyl group (C=O hydrogen-bonded) of urethane units is also identified for all samples along with the stretching vibrations related to C–H and O–H bonds at 2900 cm^−1^ and 3100 cm^−1^. The other samples highlight the same FTIR spectra results. 

#### 3.2.2. SEM Observation

The morphological structure of the produced foam samples is reported in Figure 6. The microstructure of semi-rigid foams [46] with a partially opened structure [47] is observed. The SEM micrographs highlight that the PU-Reference (Figure 6a) shows a cell dimension ranging around 400–300 µm, while the addition of the grafted CNC affects the dimension of the cell foams and the interconnection as well. In particular, the addition of pristine CNC and CNC1 induces an increase in cell size and the cell opening, as also observed by Sittinun et al. [47] in which the loading of cellulose revealed a larger and less uniform cell structure. Conversely, the addition of CNC3 leads to a decrease in cell size and an increase in interconnection within the cellular structure. This could be due to the ability of CNC3, partially exfoliated as observed through WAXS analysis, to react with isocyanate compounds. 

We could hypothesize that the CNC3 acts as a good nucleating agent in the PU foams. In particular, during the addition of CNC3 in the PU precursor, a more homogeneous and uniform mixing is observed, and it is likely that the CNC3 lowers the surface tension and the viscosity of the foaming system due to its higher availability of OH groups that interact well with the Component A (polyol and additives) and react easier with isocyanate (with respect to other functionalized and pristine CNC). This, as also observed by J.W Kang et al. [48], permits one to obtain a more uniform and finer average cell size of the PU foam.

#### 3.2.3. Contact Angle Measurements as PUFs Hydrophobicity Criterion

As reported elsewhere, the physical properties of polyurethane foam, in terms of hydrophobic characteristics of solid surface, are affected by the morphological structure and the chemical composition of solid surface. As shown in Table 5, the value of contact angle, with respect to the pristine PU foam, slightly increased as the cellulose loading from 1 wt% to 3 wt% for the PU_CNCs, PU_CNC1s, PU_CNC2s and PU_CNC4s foam samples, while a remarkable increase in contact angle from 123° to 133° for the PU_CNC1_3 and from 123° to 130° (PU_CNC3_1), 132° (PU_CNC3_2) and 137° (PU_CNC3_3) for the foams filled with CNC3 was recorded. These outcomes confirm our hypothesis that the hydroxyl groups of grafted cellulose (mainly for CNC3) reacted with isocyanate making the produced polyurethane much more hydrophobic than the pristine ones.

#### 3.2.4. Thermal Stability of PUs Prepared with Developed Fillers

The thermal stability of PU foam samples was characterized by TGA analysis. In Figure 7, the TGA of selected samples (PU-Ref, PU_CNC, PU_CNC1_3, PU_CNC2_3, PU_CNC3_3, PU_CNC4_3) is shown. In general, a polyurethane foam, characterized in an air atmosphere, highlights three main degradation steps: the first occurs around 120–200 °C which is correlated to the oxidation of the urethane segments to produce regenerated polyol, CO_2_, H_2_O species. The second degradation step, due to the oxidation of regenerate polyol, set off in the range of 200–300 °C, generates char and H_2_O, CO, CH_4_ and CO_2_ gases, and finally, the third step occurs in the 300–400 °C range in which the char is oxidized in gases [49,50,51,52]. By analyzing the TGA graph, the addition of cellulose filler positively affects the thermal degradation of polyurethane foams in all degradation ranges analyzed. For instance, with respect to polyurethane reference, an improvement in thermal stability, mainly for the PU_CNC3_3, (increasing around 10 °C in the first and the last degradation steps and 25 °C for the second degradation step) is observed. These outcomes are related to the presence of cellulose filler that brings to an increase in thermal stability of composite polyurethane foams, due to the presence of the glucosidic species [53].

#### 3.2.5. Density and Elastic Modulus of PUFs Prepared with the CNC Modified Fillers

Figure 8 reports the curves obtained from compression tests. Each of them is the average curve calculated starting from the curves of each specimen of every obtained material. As can be seen from Figure 6 and from Table 6, for each set of samples, the higher the content of the chosen filler, the higher the mechanical properties with respect to the PU_Ref. Moreover, for the sample PU_CNC3_3, the highest increase in mechanical properties is observed (see Figure 8d,f). This outcome can be ascribed to the presence of the more highly deformed and wide-spread CNC structure upon modification with TCPS (CNC3), as also stated with WAXS analysis. Moreover, to corroborate this assertion, TGA exhibited higher thermal stability for the same system, which can be associated, from a mechanical point of view, with a more intercalated CNC within the polyurethane foam. Such a configuration facilitates the gripping between the filler and the matrix, thus giving a rise to mechanical properties. Conversely, for PU samples containing CNC4 filler, the values of the modulus and of the stress at 10% of deformation are quite similar to those of the reference foam. 

All these outcomes can be attributed to the different interactions occurring between the different monomers grafted on the surface of the modified CNC and the polymer matrix, and, consequently, on how the stress is distributed between the matrix and the filler. 

As described in Section 3.1, for each set of samples, differently pre-treated CNCs were used. Each of the new grafted groups has a different chemical behavior, as well as a different tail, each interacting in a specific manner with the CNC and with polyol, and consequently with the polymer matrix.

The first important outcome stands in the alteration of the mechanical behavior of the PU foam prepared with CNC, CNC1 and CNC3 with respect to the pristine one. In detail, by increasing the amount of the filler, the PU foam changes its performance from elastomeric to elastic–plastic, with the appearance of a different kind of plateau (from the one typical of the elastic buckling to the one typical of the plastic yielding) [54]. This effect is less evident for the sample set of CNC2 and is totally absent for the CNC4 one.

This behavior can be attributed to the stiffening effect of the CNC filler in the cell edges and walls. In particular, for those samples where the used fillers better interact with the matrix, the shape of the stress–strain curve resulted deviated from one of the reference samples. 

Moreover, the sample sets with CNC, CNC1 and CNC2 experienced a quasi-linear increase in the mechanical properties while increasing the amount of the chosen filler. Instead, for those specimens where CNC3 and CNC4 were used, a different behavior was reported. For the CNC4 samples set, the shape of the curves is almost superimposable on the reference one, while for the CNC3 one, this observation can be done only for PU_CNC3_1. For PU_CNC3_2, a slight increase in the mechanical properties was observed, where there is a more notable increase in PU_CNC3_3.

To explain such behavior, a more in-depth analysis of the foam itself had to be performed by the modified version of the well-known Gibson–Ashby model described in Section 2 of the Experimental Details section.

As reported, by means of Equations (2) and (3), the mechanical properties of the foam can be directly related to the structure of the foam itself, by considering the density of the foam, the mechanical properties and the density of the material constituting the cell walls and edges. In Table 4, the values of the two ratios ρ/ρ_s_ and E/E_s_ together with exponent n and volume fraction ϕ, obtained via Equations (2) and (4) are reported. 

The Gibson–Ashby model is often used to describe and correlate the mechanical behavior of foam with the relative density of the foam itself. In most cases, a good best fit is given by Equation (3). However, in some cases, as also reported in Goods et al. [55], for polyurethane foam composites in which particles are dispersed within the polymer matrix, the exponent n in Equation (4) can result to be lower than 2, meaning that the structure of the pores is much more similar to the closed than to the open ones.

For our systems, these behavioral results are also more evident. As displayed in Figure 8, by increasing the amount of the fillers within matrix, the exponent n becomes lower. Moreover, the shape of the pores could not be modeled by a simple cubic shape, as reported by the Gibson–Ashby model, but it should be considered as a tetrakaidecahedral shape, a more complex structure that better models the cells of the foam, as also stated in Sullivan et al. [56]. In this representation, usually, the cell walls are considered to be thin enough that the material can be considered as mainly condensed in the cell edges, which will be the bearing column and beams of the foam structure. However, also, such a model does not reflect the shape of the pores observed with the SEM analyses, where the material volume is distributed among the edges and the walls of the pores. Thanks to the model introduced here, it was possible, by means of the value of the parameters ϕ and n, to quantify the amount of material distributed between the edges and walls, (see Table 5).

Table 7 reports the values of the exponential parameter n and of ϕ obtained for each sample. The parameters obtained from samples with the same percentage of filler are grouped and the average value is replaced in Equation (5). The results are reported as best fits in Figure 7.

The representative broken lines are built up considering all the samples with the same filler content. As can be observed from Figure 9, the line obtained with the model changes position depending on the selected exponent n and coefficient ϕ, depending on the filler content (the lower it is, the closer it gets to the reference curve, as expected).

## 4. Conclusions

In this paper, a protocol for the synthesis of CNC-grafted-biopolyol to be used as a successful active filler in bio-PUFs was assessed. Four different alkyl silanes were used as efficient coupling agents for the grafting of CNC and bio-polyols. 

Best results in terms of mmol of silane grafted on the CNC were obtained by using trichloro(propyl)silane (TCPS) with dried CNC and DMF as reaction medium.

Moreover, the grafting of TCPS induces a significant change in the crystalline structure of CNC (CNC3), while TMPS (CNC1) and TMOS (CNC2) and APTES (CNC4) do not seem to have any effect. This behavior could be correlated to the intercalation of TCPS within the structure of cellulose. In particular, the intercalation is higher for TCPS with respect to the more sterically hindered TMPS. This promotes a partial exfoliation of the fibrils, which became available, through OH groups, to react with isocyanate groups.

The functionalized and un-functionalized CNCs are used as reactive filler in polyurethane foams and their effects on morphological, mechanical, thermal and functional properties are determined. Morphological analysis reveals that, with respect to the PU-Ref, the CNC3 leads to a reduction of foam-cells of the final foams (PU_CNC3); on the other hand, any changes in microstructure are observed for the other PU composites. In addition, the PU_CNC3 system (mainly for the PU_CNC3_3) highlights an increase up to more than 300% in the compressive strengths and of the Young modulus, along with an improvement of thermal stability if compared to the reference ones, PU-Ref, and the PU_CNC. 

Thanks to the modified Gibson–Ashby model, by means of the mechanical properties and, as a consequence, by means of the parameters used for the model, n and ϕ were also possible to monitor the changes in the foam cells, while the filler content changed. In this view, PU_CNC3_3 exhibited the most evaluable change in the material distribution, with smaller thickened cells, and consequently, a more distributed material among cell edges and cell walls while keeping an open-cell configuration.

Furthermore, as the most valuable result, it is shown that with a suitable functionalization of CNC, it is possible to have a reactive filler able to tune the hydrophobicity, thermal stability morphological structure and consequently the mechanical performances of polyurethane foams with a low concentration of filler.

## Data Availability

The data presented in this study are available on request from the corresponding author.

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
