# Peer review of "Chemically Functionalized Cellulose Nanocrystals as Reactive Filler in Bio-Based Polyurethane Foams"

_polymers, 2021, doi:10.3390/polym13152556_

Round 1
Reviewer 1 Report
The work described in the paper is interesting but there are some deficiencies before acceptance, few points are –
- Please remove CNC from title of the paper. The abbreviation is described in abstract. So, its not required in title. In title again, authors says reactive filler. What do they mean by word “reactive”. Do the functionalized CNC react with the PU forms, do they form bonds? If yes, what type of bonds? It should be discussed in abstract.
- In abstract, authors state they target mainly target mechanical properties (#line 17) but it was shown in the manuscript that thermal stability and morphology of PU forms are also affected by functionalization of CNC. Please crosscheck this point. In keywords, it should be “cellulose nanocrystals”.
- In introduction, (#line 80), please do not use bulk references like [25-34], detail the references individually and expand them. Do this for all places throughout the paper. In introduction, please add 3-4 references from Polymers-MDPI Journal in field of present paper and discuss their advancement of present work with existing literature of polymers.
- In materials section, please discuss the details characteristics of CNC such as particle size, aspect ratio, morphology, shape etc. in a table. From section 2.2.1 to 2.2.4, please show the process chemistry and grafting, chemical bonds and other chemistry in form of scheme for all processes. In section 2.3, please detail a table of formulation, how much quantity of different functionalized are used must be reported. In section 2.4, authors are highly encouraged to perform HRTEM and XPS (binding energies) to justify grafting of functional groups on CNC surface.
- In result and discussion section, interpretation of results in section 3.1.1 and 3.1.2 is rather poor. Please discuss them in detail and the signals and corresponding functionality should be indicated in the figure as well.
- In Figure 7f, the sample PU_CNC3 shows higher mechanical properties. Why?
Reviewer 2 Report
Manuscript number: polymers-1314023
Title: “Chemically Functionalized Cellulose Nanocrystals- CNC as Reactive Filler in Bio-based Polyurethane Foams”
Authors: Francesca Coccia, Liudmyla Gryshchuk, Pierluigi Moimare, Ferdinando de Luca Bossa, Einav Barak-Kulbak, Letizia Verdolotti, Laura Boggioni and Giuseppe Cesare Lama
The paper concerns of Cellulose Nanocrystals, CNC used as fillers for flexible polyurethane foams. The work is related to rapidly developing polymeric materials such as polyurethane foams. The morphological properties, repair and mechanical parts of the final product were assessed. The work is interesting and the results are presented at a high professional level. However, it contains some arguable elements and experimental shortcomings. A proper explanations are needed. Comments and reservations:
- 2.3. PU foams preparation:
- Did the authors try to obtain the foam at a different index than [NCO]/[OH] = 1.05?
- Is the mixing speed of the components (component A and B) - 500 rpm too low? In the case of "hand casting", a much better homogeneity of the foam (especially flexible ones) is obtained above 2500 rpm.
- Why were higher filler percentages not used?
- Why were the fittings cast in free-rise and not into a closed mold (to obtain real densities of the fittings according to the area of application)?
- 3.1.3. WAXS characterization of CNCs-grafted-silanes,
- What is the reason for appearing the peak at ca. 7 degrees? (TCPS)
- It can be also seen that there's some peak at over 25 deg. (TMPS sample)
- The diffractograms should be recorded up to 40 degrees, so to observe peak at around 35 deg.
- Please prepare deconvolutions of peaks, calculate crystallinity degree.
- The pattern of CNC does not look like typicall cellulose I pattern. It seems there may be some cellulose II present.
- 2.2. SEM observation, “Specifically, we could hypothesize that the OH of CNC3 are more available to react with isocyanate respect to other functionalized and pristine CNC” On what basis did the authors present the given hypothesis? After SEM images, it is unrealistic.
- In the case of mechanical tests, did the Authors perform foam tensile tests?
- What is the purpose of the presented foam with a filler? What can the final material be used for?
- What is the economic aspect of the modified fillers used? Is it cost effective for flexible foams?
- There is a few of editorial and grammar mistakes, so paper needs to be improved carefully once again before another consideration process.
Final remark: In my opinion, the paper is interesting and is well prepared. In my opinion, the paper is worth recommendation for publication in Polymers after minor revision.
Round 2
Reviewer 1 Report
Paper can be accepted now.